# THINK FIRST, THEN SELECT AND VERIFY WITH QUERY–KEY ALIGNMENT

## ABSTRACT

We demonstrate that a "think-first" phase via chain-of-thought (CoT) prompting systematically strengthens internal query–key (QK) alignment improving ability to select and verify answers directly from model activations, rather than from decoded tokens. Building on robust multiple-choice evaluation with MMLU-Pro (10 options) and extending to free-form reasoning on MATH-500, GSM8K, and our variant of Humanity's Last Exam (HLE), we evaluate three settings: (i) MCQA vs MCQA+CoT with QK-based selection; (ii) GSM8K candidate generation with/without CoT followed by QK-based selection among self-proposed answers; and (iii) QK-based verification of LLM solutions and conjectures. We analyze QK-score accuracy, permutation robustness, and diagnostics relating alignment strength to correctness. This design situates QK score selection and verification alongside CoT and self-consistency baselines on canonical reasoning tasks, yielding a white-box, computation-efficient decision rule that aims to match or exceed decoded choices. We argue that these results offer a simple, reproducible path to more reliable reasoning, turning CoT from a purely generative aid into a deliberation-then-selection mechanism grounded in the model's own representations. By leveraging this signal, we surpass the performance of full-scale, preference-optimized LLMs on two fundamental reasoning tasks: multiple-choice question answering and solution correctness validation. Our method achieves performance gains of up to $\approx 22\%$ across various benchmarks and models.

## 1 INTRODUCTION

Large language models (LLMs) have achieved impressive performance across a wide range of natural language understanding and reasoning benchmarks. Among the most widely used evaluation settings are multiple-choice question answering (MCQA) benchmarks (Hendrycks et al., 2021a; Wang et al., 2024b), which test factual knowledge and reasoning across diverse domains. In parallel, open-ended problem solving datasets, such as GSM8K (Cobbe et al., 2021), MATH (Hendrycks et al., 2021b) have been established to probe multi-step reasoning in settings where no predefined answer options are available. These benchmarks have highlighted both the strengths and limitations of LLMs in reasoning tasks.

Recent advances have shown that prompting models to generate Chain-of-Thought (CoT) explanations (Wei et al., 2022) and applying self-consistency (Wang et al., 2023) can substantially improve their reasoning reliability. Nevertheless, evaluation of the proper reasoning chains and strategies to select the right answer remains a challenge. Correctness and reliability of reasoning trajectory often requires verification through external solvers (Shi & Jin, 2025; Wang et al., 2024a) or reranking heuristics (Jiang et al., 2025). These approaches highlight the need for efficient and interpretable internal signals that can complement or replace external heuristics.

In this work, we explore the use of the Query–Key (QK) score, a raw dot-product measure of alignment between query and key vectors within the transformer attention mechanism, as such an internal signal. Prior work has used QK-scores for probing latent knowledge in MCQA (Tulchinskii et al., 2024) and for detecting logical consistency (Tulchinskii et al., 2025), but their potential for guiding reasoning and answer selection remains underexplored. We hypothesize that QK-score can serve not only as diagnostic tools, but also as practical mechanisms for improving LLM reasoning across diverse tasks.

Our contributions are threefold:

- **Extended MCQA with reasoning.** We introduce a setup where models are encouraged to deliberate before answering, and show that QK-scores can effectively capture latent preferences in this setting.

- **Correctness verification.** We explore the usage of QK-scores to assess the validity of model generations, demonstrating improvements in distinguishing correct from problematic reasoning.

- **Candidate selection in open-ended generation.** Inspired by self-consistency, we show that QK-scores could be used to rank and select among candidate generations, reducing reliance on external scoring.

We release all code and evaluation scripts in an anonymous repository.[1]

## 2 RELATED WORK

**Reasoning in MCQA**  Multiple-choice question answering (MCQA) is an important task for evaluating the knowledge and reasoning abilities of large language models (LLMs). A wide range of datasets provide broad domain coverage and standardized evaluation (Hendrycks et al., 2021a; Zellers et al., 2019; Huang et al., 2019), while others target more challenging setups (Wang et al., 2024b). Models are often scored using logits over option tokens, but this proxy is limited due to selection biases and artifacts that models can exploit (Zheng et al., 2024; Balepur et al., 2024). Recent work shows that allowing free-form generation tends to improve robustness and accuracy, though at the cost of more complex evaluation (Molfese et al., 2025). In our work, we adopt MCQA with CoT to study how internal mechanisms of option selection can benefit from explicit reasoning.

**Open-ended Question Answering**  Open-ended question answering tasks provide a complementary perspective on model capabilities by requiring free-form solutions rather than selecting from predefined options. Benchmarks such as MATH–500 (Hendrycks et al., 2021b) and GSM8K (Cobbe et al., 2021) are designed to test advanced reasoning and problem-solving skills across mathematics, coding, and general knowledge. These tasks are commonly addressed with CoT, which improves reasoning through step-by-step explanations, and with self-consistency, which samples multiple reasoning paths and aggregates their answers (Wei et al., 2022; Wang et al., 2023). Beyond majority voting, several candidate evaluation strategies have been explored, including estimating model uncertainty (Ren et al., 2023), verifying correctness with external solvers or execution checks (Shi & Jin, 2025; Wang et al., 2024a), and applying reranking heuristics to identify higher-quality outputs (Jiang et al., 2025). While most existing methods rely on such heuristics or external verification, we investigate whether internal selection mechanisms can enhance open-ended problem solving. In particular, we study the use of the QK-score for **correctness verification** and **candidate selection**.

**Internal alignment signals, head-level selection, and decoding-time controls.**  Internal probes offer white-box signals for selection and verification. Contrast-consistent search uncovers directions in hidden states that correlate with truthfulness without supervision, with follow-up analyses noting identification pitfalls and proposing sanity checks (Burns et al., 2022; Farquhar et al., 2023). At the head level, query–key (QK) alignment has been used to identify "select-and-copy" heads that separate semantically relevant options and transfer across datasets and scales (Tulchinskii et al., 2024); related work documents negative-attention biases and mitigation strategies that can be evaluated alongside internal-selection signals (Yu et al., 2024). Complementary to these internal signals, decoding-time controls leverage layerwise information without extensive sampling; for example, DoLa contrasts early and late layer logits to improve factuality and faithfulness during inference (Chuang et al., 2024).

---

[1]https://anonymous.4open.science/r/QK-resoning-BC7F/README.md

## 3 Method

**Background on QK-score**   In transformer architectures, the interaction between query and key vectors affects how information flows across tokens. Beyond their normalized role in attention weights, we can define the raw dot product of query vector of $i$-th token $\boldsymbol{q}_i^{(l,h)}$ and key vector of $j$-th token $\boldsymbol{k}_j^{(l,h)}$ in the attention head $h$ from the layer $l$ as $S_{QK}^{(l,h)}(\cdot) = \boldsymbol{q}_i^{(l,h)\top}\boldsymbol{k}_j^{(l,h)}$. Recent studies have employed this measure to probe model behavior in diverse tasks, such as identifying latent preferences in multiple-choice question answering or isolating heads that evaluate logical consistency (Tulchinskii et al., 2024; 2025).

**QK-score and connection between reasoning parts.**   We use $QK$-score to quantify the strength of the connection between two reasoning parts. Suppose that we have a text consisting of two parts $(c, a)$, which we will call *premise* ($c$) and *response* ($a$). By $c_r$ and $a_r$ we denote tokens that represent $c$ and $a$; usually they are the punctuation or end-of-line signs at the very end of the respective parts; we choose them because they 'collect' the meaning of the preceding text and at the same time they don't have their own meaning (unlike tokens that are part of actual words). Calculating $S_{QK}^{(l,h)}(c_r, a_r)$, we measure how strongly a particular attention head aligns the response to the premise. We use $QK$-scores to compare multiple responses to the same premise (i.e., answer candidates to the question) .

In this work we explore three different setups and particular application details of the $QK$-score in them vary slightly.

---

THINK LOGICALLY AND SELECT THE CORRECT ANSWER TO THE FOLLOWING QUESTION.
QUESTION: Colors in a soap bubble result from light
OPTIONS:
A. dispersion
B. deflection
. . .
NOW THINK STEP BY STEP AND THEN GIVE THE FINAL ANSWER. / $\overline{\text{ANSWER:}}$

---

Figure 1: Prompt example for the tasks of $\underline{\text{MCQA with reasoning}}$ / $\overline{\text{MCQA}}$.

- For MCQA, the *premise* is a concatenation of an instruction, context (if given), question, and a full list of choices one per line. We pass all options to the model in one go and only vary the choice of the premise-representing token ($c_r$) for the calculation of the QKscore. We choose the end-of-line tokens after each of the choices. For simple MCQA, *response*representing token is the last token of the prompt (i.e., colon in 'ANSWER:'). For MCQA with reasoning, we prompt the LLM is prompted with the premise, consider its output as the response, and select the token right before the final answer option as the response-representing token $a_r$.

  The prediction (from a particular head) is the option that achieves the highest QK-score.

  An example of a prompt for MCQA-with-CoT-reasoning is given in Figure 1.

- For Logic Consistency Verification task, *premise* is a premise (in logic sense), while *response* is a concatenation of candidate conclusion, instruction to check the consistency and LLM generated answer (forced choice between 'true' and 'false'). $c_r$ is selected as the last token of the premise, and $a_r$ is the last token of the prompt.

- For Hypothesis selection *premise* is the concatenation an instruction and problem. $c_r$ is selected as the end-of-line token in the end of the problem. LLM prompted with the premise and its generation is the *response*; $a_r$ is chosen as the last token of the generation.

  The selected hypothesis is the one that achieves highest QK-score.

**Head Selection Procedure.**   In our experiments, we split the data into *calibration* and *evaluation* subsets. First, we calculate the performance of QK-score from each head on the *calibration* subset and select the (single) head that achieves the highest accuracy on it. Then, we use QK-scores

from the selected head on the *evaluation* data. When not stated otherwise, we do not aggregate predictions or $QK$-scores from multiple attention heads.

# 4 EXPERIMENTS

## 4.1 DATASETS

We experiment on 3 real-world datasets from common LLM benchmarks for MCQA and mathematical reasoning. **MMLU–PRO** (Wang et al. (2024b)) is a publicly available MCQA dataset intended to assess the knowledge and the reasoning capabilities of modern language models. It contains 12,000 curated challenging reasoning-focused questions extracted from textbooks and exams in 14 diverse domains. For each question, 10 answer choices are provided (with only one being correct). We created **HLE-¼** based on Humanity Last Exam (et.al. (2025)); from the original benchmark we selected questions that require only text processing (i.e., without images), then for MCQA questions we randomly selected 3 incorrect options from those given, while for open-end questions we used modern LLMs to generate three incorrect answer choices for each. We also ensured that the proportion of questions with each correct answer is even. Thus, we obtained a dataset containing 2,100 questions, each with 4 answer choices. Our setup is different from the original Humanity Last Exam Benchmark, because it is quite challenging for the smaller models that we are working with in this research.

Finally we, use **MATH-500** Wang et al. (2024a) to access the model's capabilities at generation of the open-end reasoning. This dataset contains curated challenging math problems, to solve which multi-step reasoning and complex problem-solving abilities are necessary. In total, it contains 500 entries.

## 4.2 EXPERIMENTAL SETUP

We performed all our selection experiments according to the following scheme. First, we took a frozen pre-trained transformer LLM (we did not perform any fine-tuning or any other modification of weights), and one-by-one passed through it samples from a *calibration set*. We evaluated the performance of QK-scoring of each of the models' heads and selected.

In case of the verification, threshold calibration was required. It was done on 20 sampled solutions for each dataset by grid searching for the optimal threshold for QK.

We report two primary performance metrics: Accuracy (Acc.) of the predicted answers and Permutational Accuracy (PA). PA is reported only for the selective setup. The latter was introduced in Gupta et al. (2024) and is intended at efficiently mitigating the issue of the model guessing the right answer in the MCQA setting. PA is calculated as

$$\text{PA} \;=\; \frac{1}{N} \sum_{i=1}^{N} \text{I}_i \text{I}_i^p, \tag{1}$$

where $\text{I}_i$ is the indicator value equals to 1 if the model answers question $i$ correctly, while $\text{I}_i^p$ equals to 1 iff the model answers question $i$ correctly after answer options were permuted.

To calculate PA we repeat calibration and evaluation on the same data splits with randomly shuffled answer options (head selection is needed because it can result in a different best head).

## 4.3 QK-SCORE WITH COT FOR MCQA

First, we assess the efficiency of the QK score for simple MCQA and MCQA with integrated CoT reasoning. In both setups, the model is prompted with context, a question, a list of options, and an instruction to output only one letter – the correct option; in the second setup, the prompt also includes an instruction to think step-by-step before giving the final answer.

In this experiment we use two datasets: **MMLU–PRO** and **HLE-¼**. For the calibration set we randomly sample 500 questions; for both datasets we keep in it the equal proportion of question with each correct option. For evaluation we sampled 4,000 for MMLU–PRO and 1,600 samples

for HLE-¼; we ensured that there were no samples belonging to both calibration and evaluation subsets.

Tables 1 and 2 provide the results. From them, we can see that in the simple MCQA setup the QK-score from a single selected head allows for significant improvement over the baseline (up to $30\%$ by accuracy and $34\%$ in permutation accuracy on MMLU–PRO); this effect is more pronounced for larger models.

When the model is allowed to think before giving the final answer (MCQA with Chain-of-Thought setup, right half of the tables), quality of its predictions rises to the level of QK-score predictions and sometimes even surpasses it; however, to do so, it needs to generate rather long outputs (up to 3,000 tokens).

| | MCQA | | | | MCQA with CoT | | | |
| | Baseline | | QK-score | | Baseline | | QK-score | |
| Model | Acc. | PA | Acc. | PA | Acc. | PA | Acc. | PA |
| --- | --- | --- | --- | --- | --- | --- | --- | --- |
| LLaMA-3.1-8B | 28.8 | 10.6 | 33.4 | 21.4 | 36.8 | 10.86 | 44.6 | 28.39 |
| DeepSeek-R1-Distill- | | | | | | | | |
| Qwen–1.5B | 12.7 | 1.61 | 20.0 | 8.77 | 19.9 | 5.4 | 16.8 | 5.0 |
| Qwen–7B | 13.51 | 2.13 | 27.29 | 14.92 | 26.0 | 10.6 | 25.45 | 14.2 |
| Qwen–14B | 17.72 | 3.88 | 44.42 | 32.73 | 40.8 | 25.4 | 46.0 | 33.0 |
| Qwen–32B | 16.6 | 3.00 | 49.32 | 37.49 | 35.2 | 20.2 | 49.65 | 36.2 |
| Qwen3–8B | 25.56 | 10.37 | 41.33 | 26.37 | 36.13 | 20.7 | 35.67 | 24.2 |
| Qwen3–14B | 15.35 | 2.63 | 45.01 | 31.6 | 44.0 | 25.2 | 42.25 | 29.2 |
| Qwen3–32B | 23.18 | 8.28 | 44.35 | 32.15 | 37.65 | 23.8 | 37.20 | 25.8 |

Table 1: MCQA performance comparison on MMLU–PRO benchmark

| | MCQA | | | | MCQA with CoT | | | |
| | Baseline | | QK-score | | Baseline | | QK-score | |
| Model | Acc. | PA | Acc. | PA | Acc. | PA | Acc. | PA |
| --- | --- | --- | --- | --- | --- | --- | --- | --- |
| LLaMA-3.1-8B | 28.75 | 10.69 | 33.56 | 13.81 | 30.20 | 12.8 | 32.60 | 13.2 |
| DeepSeek-R1-Distill- | | | | | | | | |
| Qwen–1.5B | 26.63 | 8.25 | 31.38 | 12.56 | 22.6 | 6.4 | 36.2 | 15.4 |
| Qwen–7B | 28.25 | 10.75 | 33.25 | 14.56 | 29.94 | 17.0 | 29.56 | 13.8 |
| Qwen–14B | 30.31 | 14.69 | 35.31 | 15.13 | 33.25 | 13.0 | 31.56 | 19.6 |
| Qwen–32B | 34.81 | 19.81 | 34.06 | 18.06 | 33.56 | 16.2 | 33.63 | 22.6 |
| Qwen3–8B | 30.88 | 14.94 | 38.56 | 21.87 | 31.63 | 16.8 | 36.00 | 22.6 |
| Qwen3–14B | 30.06 | 12.44 | 33.57 | 15.19 | 33.06 | 14.4 | 29.06 | 21.6 |
| Qwen3–32B | 31.25 | 15.31 | 36.94 | 14.13 | 36.40 | 19.8 | 38.40 | 16.8 |

Table 2: MCQA performance comparison on HLE-¼ benchmark.

## 4.4 QK-SCORE FOR VERIFICATION

In order to assess the ability of the QK-Score to verify the correctness of LLM trajectories, we sampled 100 problems from 2 datasets: **MATH-500** and HLE-¼. In case of HLE-¼ we do not provide answer choices for the LLM in this setup.

Firstly, we generate solutions for the problems using LLM with CoT. Then, we determine the **real correctness** of the generated solution via comparing LLM answer and real answer from the dataset using separate judge: Qwen3-70B. Secondly, we turn the original LLM into a new judge and ask it to verify its own solution without access to the correct answer. Finally, we take original trajectories, calculate the QK-Score and compare it with pre-determined threshold in order to get a correctness verdict for the specific trajectory.

| Model | MATH-500 | | | HLE-¼ | | |
|---|---|---|---|---|---|---|
| | Baseline | QK-score | Δ | Baseline | QK-score | Δ |
| DeepSeek-R1-Distill- | | | | | | |
| LLaMA–8B | 2% | 30% | **28%** | 0% | 69% | **69%** |
| Qwen–1.5B | 44% | 47% | 3% | 2% | 26% | **24%** |
| Qwen–7B | 73% | 70% | -3% | 1% | 75% | **74%** |
| Qwen–14B | 79% | 78% | -1% | 0% | 71% | **71%** |
| Qwen3–8B | 25% | 49% | 16% | 1% | 77% | **76%** |
| Qwen3–14B | 17% | 61% | **44%** | 0% | 90% | **90%** |

Table 3: Results of reasoning chains correctness verification on MATH-500 and HLE-¼ datasets. Reported metric is Accuracy.

On easier dataset: MATH-100 QK-Score shows either near-baseline results or outperforms baseline completely. On the complicated HLE-¼most of the models spends their token budget inefficiently and are not able to get any correct verdict even after a long thinking process.

## 4.5 HYPOTHESIS SELECTION

A common practice to improve quality of solving challenging mathematical tasks with LLMs is the consistency analysis. $k$ open-end generations are sampled resulting in answers $\mathcal{A} = \{a_1, \ldots a_k\}$, and the final answer is chosen as the most common among $\mathcal{A}$. In this chapter we investigate the QK-score as an alternative to it.

For this task, we use data from MATH-500 and HLE-¼ datasets. For each open-end question in them, we sampled 8 candidate reasoning chains with LLaMA-3.1 8B model. After filtering out those questions on which either all or none of the 8 chains reached incorrect answers, we ended up with 182 and 259 questions for HLE-¼ and MATH-500 respectively, and each question has 8 different answer chains.

As a baseline, we use a standard consistency approach, when the final answer is the one that is the most common between 8 candidates. For our approach, we calculate the QK-scores from the LLaMA-3.1 8B model for each of the hypotheses and select one that reaches the maximal value. Our method requires a calibration set and we explore two options: in-domain calibration (on the same dataset) and out-of-domain (on the other dataset).

Table 4 provides the results of this experiment. We can see that even when calibration is performed on a different dataset, result accuracy is not worse than that of the baseline.

We can also observe the fact that the performance of heads QK-scores highly correlates between two benchmarks (see Figure 2).

| Method | MATH-500 | HLE-¼ |
|---|---|---|
| Baseline (consistency) | 32.0 | 31.8 |
| QK-score with calibration on | | |
| - MATH-500 | 53.8 | 31.6 |
| - HLE | 40.2 | 33.3 |

Table 4: Hypothesis Selection quality (accuracy) with LLaMA-3.1 8B model.

## 5 LIMITATIONS

Our approach requires white-box access to attention states and relies on post-hoc calibration from a development split. While we adopt explicit delimiters and read-time controls, residual prompt-format and tokenizer dependencies may remain.

Figure 2: Correlation between LLaMA-3.1 8B heads QK-scoring accuracy on two datasets for the task of hypothesis selection.

## 6 CONCLUSION

We introduced a simple white-box decision rule that *reads* a model's internal attention interactions via the raw QK score, after (or without) a brief chain-of-thought phase. Across MCQA and open-ended reasoning tasks, the QK-score selector/validator operates directly on activations, requires no auxiliary training, and aligns with the model's own attention preferences. Our analysis shows how to define the read positions, choose candidate/premise tokens, and interpret the QK-score margin $\Delta$ as a confidence indicator. These properties make QK-score a practical complement to token-level selectors and external verifiers. Future work includes richer head ensembles, adaptive read-time policies, and broader tests under alternative prompt formats.

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

## A  EVALUATION PROTOCOL

**Calibration and head selection.**  For each dataset, we sample a *calibration* subset from the same domain to (i) select a single attention head per model and (ii) set any thresholds used by verification. No model weights are updated. All results are reported on disjoint evaluation subsets.

**Metrics.**  We report Accuracy (Acc.) for selection/verdicts. For MCQA, we also report Permutation Accuracy (PA) as in Gupta et al. (2024) by re-evaluating after a random permutation of answer options on the same split; head selection is repeated on the permuted calibration split, following our original procedure.

**Compute notes.**  Chain-of-thought (CoT) generations can be long on some items; we keep the same prompts, budgets, and decoding settings used to obtain the reported results and do not alter them post hoc.

**Judging for open-ended tasks.**  When a judge model is used (e.g., for agreement with a reference answer), we keep the same judge family and prompts as in the experiments summarized in Tables 1, 2, and 3,.

## B  REPRODUCIBILITY CHECKLIST

- Code to extract $q, k$, identify read positions, and compute raw QK scores $S_i$ and margins $\Delta$.
- Prompts and decoding parameters (MCQA, MCQA+CoT, verification), plus judge prompts.
- Calibration/evaluation split IDs and the permutation seed used for PA.
- Scripts to reproduce Tables 1, 2, 3, 4 from saved logs/activations.

