# OpenReview forum: "Think First, Then Select and Verify with Query–Key Alignment"
_ICLR.cc/2026/Conference — Submitted to ICLR 2026_

### Official Review · Reviewer_vAFz · 2025-10-16

**Soundness:** 1
**Presentation:** 1
**Contribution:** 1
**Rating:** 0
**Confidence:** 5

**Summary:**

This paper proposes using internal QK-scores (attention scores without RoPE + before softmax) for
1. MCQA Evaluation (Section 4.3): selecting the model’s choice during Multiple Choice QA evaluation (instead of generating answer token)
2. Self-Verification (Section 4.4): verifying model’s own reasoning step
3. Candidate Generation Selection (Section 4.5): selecting the most promising generation (instead of majority voting)

While previous works (both cited in the paper) have explored the use of QK-scores for MCQA without CoT [1] and Self-Verification [2], the paper hopes to extend the setting to MCQA with CoT and Candidate Generation Selection. The extension from MCQA without CoT to MCQA with CoT is barely incremental. The paper claims that they have novelty beyond [2], but it is unclear how their setting on Self-Verification is different. The only novel contribution, if any, would be the Candidate Generation Selection. However, this setting can also be understood as a variation of an MCQA (the question is "which generation is most promising" and each option is the candidate generation).

[1] Tulchinskii, Eduard, et al. "Listening to the Wise Few: Select-and-Copy Attention Heads for Multiple-Choice QA." arXiv preprint arXiv:2410.02343 (2024).

[2] Tulchinskii, Eduard, et al. "Quantifying Logical Consistency in Transformers via Query-Key Alignment." arXiv preprint arXiv:2502.17017 (2025).

**Strengths:**

I find no strength in this paper. Instead, I will elaborate on its weaknesses.

**Weaknesses:**

# Lack of Novelty/Contribution
Using QK-values to choose the best option in Multiple Choice QA evaluation has already been proposed [1]. The only addition this paper proposes is the usage of chain-of-thought (CoT). This is barely incremental. Furthermore, the paper later states that with CoT, the baseline approach (letting model generate the answer token) catches upto the performance of their proposed QK-value-based method (line 193, page 4). This further downweights the significance of their results.

Using QK-values to verify the model's own answer has already been proposed [2]. It is unclear how the proposed setting in this paper is any different from [2].

Selecting the most promising generation can be understood as a Multiple Choice QA, as is currently written. The question would be "what is the most promising generation" and the list of options would be each of the candidate generations.

[1] Tulchinskii, Eduard, et al. "Listening to the Wise Few: Select-and-Copy Attention Heads for Multiple-Choice QA." arXiv preprint arXiv:2410.02343 (2024).

[2] Tulchinskii, Eduard, et al. "Quantifying Logical Consistency in Transformers via Query-Key Alignment." arXiv preprint arXiv:2502.17017 (2025).

# Lack of Details of Experimental Setup
The paper is lacking details that are fundamental to understand the experimental setup.
1. The prompt examples for Section 4.4 and Section 4.5
2. Whether the MCQA evalution is done zero-shot (Section 4.3)
3. The decoding setting (temperature, top-p etc.)
4. How exactly is the best QK head selection from the validation set?
5. Why do you need an external LLM-as-a-judge (Qwen3-70B) (line 247)? The answer for MATH-500 should be standardized right? Is is just for HLE? What is the prompt for this external LLM-as-a-judge?

etc. etc.

Many of the terms are not used consistently throughout the paper and add to the confusion. For example, MCQA+CoT (line 16), MCQA with CoT (line 67, 196, 216), MCQA with reasoning (line 51, 118, 124), MCQA-with-CoT-reasoning (line 128), MCQA with integrated CoT reasoning (line 181) all seem to refer to the same thing.

# Experimental Setup Copied From Previous Works?
Some parts of the experimental setup are either out-of-place or follow too closely to the two previous work s [1, 2], but do not properly acknowledge this resemblance.

The term "premise" in the paragraph under **QK-score and connection between reasoning parts.** (lines 100-1345) seems out of place, potentially except for the Logic Consistency Verification (line 129). It is also unclear why "premise" would be abbreviated as "c" (line 102). Upon further inspection, it seems like this term was directly copied from [2]. Relevant part from [2]: "In our setup, each input consists of a context c (which provides the premises), a statement s (a candidate conclusion), and a candidate answer a_i" (page 2).

The paper mentions: "When it is not stated otherwise, we do not aggregate predictions or QK-scores from
multiple attention heads. Instead, in each experiment we use a separate calibration subset of the data from the same domain to select the single best performing head." (line 137-139). This is weird since the paper never mentions aggregating scores acros attention heads. Upon inspection, this seems to be modified from a similar sentence from [1]: "We do not aggregate heads predictions. Instead, we use the scores from the single best head, which is selected by the accuracy on the validation set D_val." (page 4).

The paper defines Permutational Accuracy (PA) in Equation 1 and explains "where I_i is the indicator value equals to 1 if the model answers question i correctly, while I^p_i equals to 1 iff the model answers question i correctly after answer options were permuted." (lines 174-175). This is almost word-to-word copied from [1]: "where I_i is the indicator value equals to 1 iff model’s answer on question i is correct, while I^p_i equals to 1 iff model gives correct answer on question i after its options (their texts not letters) were permuted" (page 6).

None of these similarities are properly acknowledged.

[1] Tulchinskii, Eduard, et al. "Listening to the Wise Few: Select-and-Copy Attention Heads for Multiple-Choice QA." arXiv preprint arXiv:2410.02343 (2024).

[2] Tulchinskii, Eduard, et al. "Quantifying Logical Consistency in Transformers via Query-Key Alignment." arXiv preprint arXiv:2502.17017 (2025).

# Questionable Results
The baseline numbers for MCQA (just letting model generate answer tokens) seem off. See Table 1 for example.
1. DeepSeek-R1-Distill models and Qwen3 models have lower / similar numbers than LLaMA-3.1-8B?
2. No benefit of model scale? For example, 8B/14B/32B models do not show a smooth increase in performance?
3. Qwen3 numbers are significantly lower than expected. For example, Qwen3-32B should have 68% on MMLU-Pro (5-shot) [3]. Even if this paper used zero-shot and used a different evalution setting (which is why the paper should have added more details on what setting they used), the number should not change this much.

[3] Yang, An, et al. "Qwen3 technical report." arXiv preprint arXiv:2505.09388 (2025).

# Internally Inconsistent Content
Now this is getting into the interesting part. Upon manual inspection of the codebase provided by the authors, not only do they not provide the full codebase, some of their results do not match the content shown in the paper.

For example, in the 16th and 17th output cell of the `HLE_MCQA_qwen3_14b.ipynb` file in the codebase, they report "Baseline: 0.336 QK: 0.354" and "Baseline: 0.36 QK: 0.35", both of which numbers are not consistent with Table 2.

For another example, in Figure 1, the paper claims that they use "Options:\n" in the prompt for MCQA. However, in the provided `HLE_MCQA_qwen3_14b.ipynb` file in the codebase, such text is not included.

# Numerous Mistakes in the References Section
There are many mistakes in the References section that raise suspicision. These mistakes are not typical mistakes made by humans.


1. The paper writes "DoLa: Decoding by contrasting layers improves **factuality and faithfulness**" (line 334-335). The correct title of this paper is "DoLa: Decoding by Contrasting Layers Improves **Factuality in Large Language Models**"
2. The paper includes "Decoding-time baseline." (line 335) at the end of an entry under Chuang et al. The paper also includes "Introduces the GSM8K benchmark." (line 338) at the end of an entry under Cobbe et al. These seem to be comments that humans generally won't include.
3. For the entry under Ren et al. (lines 367-368), the name of the journal is simply "Proceedings on"

# Nitpicky Details
1. The paper changed the margin setting. Left margin is 1inch? But this is interesting since the paper will still fit under the 9 page limit with the permitted margin setting.
2. No description of the green color in Table 3.
3. 16% in Table 3 is not colored green. It also should be 14% instead.

**Questions:**

1. Can you provide more details into the experimental setup? Specifically, the prompts for Section 4.4, Section 4.5 would be useful. Also, please provide the decoding budget and the temperature / top-p setting for all experiments.

2. Can you explain how the baseline numbers were derived in Table 1 and 2? Why are the numbers so different from what you might expect of the models?

3. If the Baseline approach with CoT performs as well as your QK-value-based approach, what is the significance?

4. In your abstract, you say "By leveraging this signal, we surpass the performance of full-scale,
preference-optimized LLMs on two fundamental reasoning tasks: multiple-choice question answering
and solution correctness validation." What are the "full-scale preference-optimized LLMs"?

5.  In your abstract, you say "Our method achieves performance gains of up to ≈ 22% across various benchmarks and models." Where do you get a gain of "22%"?

---

> ### Author Response · Authors · 2025-11-22
> **Rebuttal 1/4**
>
> 1. The only addition this paper proposes is the usage of chain-of-thought (CoT). This is barely incremental. Furthermore, the paper later states that with CoT, the baseline approach (letting the model generate the answer token) catches upto the performance of their proposed QK-value-based method (line 193, page 4). This further downweights the significance of their results.
>
> **Reply:** We respectfully disagree that the only addition is the use of chain-of-thought. While prior work has employed the QK signal for probing (which we acknowledge in our paper), we do not consider the generalization of this method to reasoning setups to be an incremental contribution. The existence of heads capable of providing strong reasoning signals was previously unknown, as was the mechanism for selecting them and the ability to demonstrate comparable or superior performance across various challenging setups that require reasoning abilities.
>
> 2. Using QK-values to verify the model's own answer has already been proposed [2]. It is unclear how the proposed setting in this paper is any different from [2].
>
> **Reply:** Prior work [1][2] has demonstrated the ability to use the QK signal on relatively simple setups. To be more precise, we provide an example of a task from the dataset used in [2]:
>
> ```
> Harry is strong. Harry is big. Harry
> is high. Anne is thin. Anne is little. Gary is
> smart. Gary is quiet. Gary is kind. Fiona is poor.
> Fiona is rough. Fiona is sad. Strong people are
> smart. If someone is thin and little then they are
> short. If someone is poor and rough then they
> are bad. If someone is smart and quiet then they
> are nice. All short people are small. All smart
> people are quiet. All nice people are wealthy.
> All bad people are dull.
> STATEMENT: Harry is quiet.
> ```
>
> Once again, generalization from this simple setup is non-trivial. The present paper introduces a new head selection procedure for reasoning models and demonstrates successful application of the QK signal to tasks of higher difficulty.
>
> 3. The paper is lacking details that are fundamental to understand the experimental setup.
> The prompt examples for Section 4.4 and Section 4.5
> Whether the MCQA evalution is done zero-shot (Section 4.3)
> The decoding setting (temperature, top-p etc.)
> How exactly is the best QK head selection from the validation set?
> Why do you need an external LLM-as-a-judge (Qwen3-70B) (line 247)? The answer for MATH-500 should be standardized right? Is is just for HLE? What is the prompt for this external LLM-as-a-judge?
>
> **Reply:** We thank the reviewer for the feedback. While some details of the setup are already included in the text, we will provide a more detailed description in the revised version of the present paper..
> Unfortunately, we are unable to understand the intent of question 4. If it is related to the selection procedure, we provide a comprehensive description in our paper (Section 3). Otherwise, we sincerely ask for clarification.
> For both the HLE and MATH datasets, we used an LLM-as-a-judge to compare the answers in order to maintain consistency between these two setups.
>
> 4. Some parts of the experimental setup are either out-of-place or follow too closely to the two previous works [1, 2], but do not properly acknowledge this resemblance. <...> None of these similarities are properly acknowledged.
>
> **Reply:** We respectfully disagree with the reviewer, as we believe these similarities are properly acknowledged. Both papers are cited, and neither has been published in formal venues. The use of portions of the codebase is also standard practice, as the newer results are literal improvements on the earlier results.

---

> ### Author Response · Authors · 2025-11-22
> **Rebuttal 2/4**
>
> 5.
> The baseline numbers for MCQA (just letting model generate answer tokens) seem off. See Table 1 for example.
>    1. DeepSeek-R1-Distill models and Qwen3 models have lower / similar numbers than LLaMA-3.1-8B?
>    2. No benefit of model scale? For example, 8B/14B/32B models do not show a smooth increase in performance?
>    3. Qwen3 numbers are significantly lower than expected. For example, Qwen3-32B should have 68% on MMLU-Pro (5-shot) [3]. Even if this paper used zero-shot and used a different evalution setting (which is why the paper should have added more details on what setting they used), the number should not change this much.
> [3] Yang, An, et al. "Qwen3 technical report." arXiv preprint arXiv:2505.09388 (2025).
>
> **Reply:** Thank you for raising these points. After reviewing once again our experimental setup, we identified a potential cause in difference (i.e., absence of line “Options:\n”, see below)  in our prompt design.
>
> We have following preliminary results on DeepSeek-R1-Distil-Qwen7B on standard prompt design (with line “Options:\n” included) on MMLU_Pro:
>
> | MCQA                     |             | MCQA with CoT         |             |
> |--------------------------|-------------|------------------------|-------------|
> | **Metric**               | **Baseline**| **QK-score**           | **Baseline**| **QK-score** |
> | Accuracy                 | 12.132      | 27.563                 | 26.734      | 46.327       |
> | Permutation Accuracy     | 1.725       | 14.457                 | 16.531      | 31.633       |
>
> In most of the cases there are marginal differences; the only exception is QK-score on full reasoning chains, where standard prompt results in a significant improvement in performance of the QK-score (+15%), while the improvement on the baseline is not significant (+0.7%). We will conduct additional experiments with standard design to isolate the issue, and properly document the prompt differences in the text. We will upload the result in a follow-up comment once they are ready.
>
> Regarding (3): in all of our experiments we work in 0-shot setup only; we do not use in-context examples (in our experiments on other models it often happened to cause noticeable change in performance), and there are also some minor prompt design differences.
>
> 6. Now this is getting into the interesting part. Upon manual inspection of the codebase provided by the authors, not only do they not provide the full codebase, some of their results do not match the content shown in the paper.
>
> For example, in the 16th and 17th output cell of the HLE_MCQA_qwen3_14b.ipynb file in the codebase, they report "Baseline: 0.336 QK: 0.354" and "Baseline: 0.36 QK: 0.35", both of which numbers are not consistent with Table 2.
> For another example, in Figure 1, the paper claims that they use "Options:\n" in the prompt for MCQA. However, in the provided HLE_MCQA_qwen3_14b.ipynb file in the codebase, such text is not included.
>
> **Reply:** We would like to point out that in the provided code we output performance on both calibration and evaluation subsets, while in the paper we report metrics only for the evaluation subset. Cells 16 and 17 which have outputs "Baseline: 0.336 QK: 0.354" and  "Baseline: 0.36 QK: 0.35" are part of the calibration computations (i.e., selection of the best performing head on a separate data subset), as it is explicitly stated in the notebook.  The evaluation performance is computed in Cell # 21 (titled “Test performance”), and results from there  perfectly match those reported in Table 2.
>
> In regards to “Options:\n” line. Thank you for pointing out this issue; indeed it shouldn’t be included in the listing on Figure 1, and we will fix it in the final version. It is a typo that was left from an alternative experiment design that was considered but was not included in the paper.

---

> ### Author Response · Authors · 2025-11-22
> **Rebuttal 3/4**
>
> 7. There are many mistakes in the References section that raise suspicision. These mistakes are not typical mistakes made by humans.
> The paper writes "DoLa: Decoding by contrasting layers improves factuality and faithfulness" (line 334-335). The correct title of this paper is "DoLa: Decoding by Contrasting Layers Improves Factuality in Large Language Models"
> The paper includes "Decoding-time baseline." (line 335) at the end of an entry under Chuang et al. The paper also includes "Introduces the GSM8K benchmark." (line 338) at the end of an entry under Cobbe et al. These seem to be comments that humans generally won't include.
> For the entry under Ren et al. (lines 367-368), the name of the journal is simply "Proceedings on"
>
> **Reply:** We thank the reviewer for pointing out typographical issues, as we had not thoroughly checked for typos in the references section. However, we strongly disagree with the accusation implied in this comment. Furthermore, it is important to note that we have the discretion to edit and organize the bibliography as we see fit. Drawing serious conclusions based solely on minor typos in the reference section is clearly unreasonable.
>
>
> 8. The paper changed the margin setting. Left margin is 1inch? But this is interesting since the paper will still fit under the 9 page limit with the permitted margin setting.
> No description of the green color in Table 3.
> 16% in Table 3 is not colored green. It also should be 14% instead.
>
> **Reply:** We thank the reviewer for pointing out the formatting issues. We will address them in the camera-ready version of the paper and will pay particular attention to the first issue.
>
> [1] Tulchinskii, Eduard, et al. "Listening to the Wise Few: Select-and-Copy Attention Heads for Multiple-Choice QA." arXiv preprint arXiv:2410.02343 (2024).
> [2] Tulchinskii, Eduard, et al. "Quantifying Logical Consistency in Transformers via Query-Key Alignment." arXiv preprint arXiv:2502.17017 (2025).

---

> > ### Comment · Reviewer_vAFz · 2025-11-23
> > **Please submit an updated version of the manuscript first**
> >
> > Dear Authors,
> >
> > I appreciate your response. However, I would highly suggest you upload an updated version of the manuscript, which is possible under the ICLR guidlines. The current version violates the formatting requirement (editing the margin) which technically is subject to desk rejection and is confusingly written as pointed out by not just myself but also other reviewers.
> >
> > I will take another read through the manuscript and reassess my reviews.
> >
> > Thanks

---

> ### Author Response · Authors · 2025-11-23
> **Formatting fix**
>
> We have uploaded a revision that addresses the margin issue. We sincerely appreciate the reviewer for bringing this problem to our attention. We will also address other presentation issues as soon as possible.

---

> ### Author Response · Authors · 2025-12-03
>
> Taking into account the present circumstances related to the new rebuttal format, and with full respect to the reviewer, we would like to highlight to our AC that the reviewer’s claims unrelated to presentation were either addressed or disproved. While the reviewer has put effort into evaluating the paper, many of the claims that the reviewer has provided were either unsupported or incorrect. Moreover, the reviewer explicitly agreed to fully reassess the review, but due to the constraints of the new rebuttal format, this is no longer possible. Therefore, we feel that the rating provided by the reviewer is not well-supported, and given that the reviewer himself suggested reassessing the review from scratch, we believe that this rating should not be taken into account in the final decision.

---

### Official Review · Reviewer_ZnYX · 2025-10-25

**Soundness:** 1
**Presentation:** 1
**Contribution:** 1
**Rating:** 2
**Confidence:** 3

**Summary:**

This paper is not written well enough for me to be able to understand it clearly and appreciate the contributions. Based on what little I could make out, this paper is trying to solve MCQA task by forcing model to do CoT before selecting final answer choice. And, in that process, they are aiming to use QK scores which is part of self-attention calculation anyways. It is unclear to me what is novel idea in this paper and why standard self-attention mechanism can’t handle what is being proposed here. I am neither very clear about the research gap that is being addressed in this paper nor the novelty of the contributions. At the least, this paper needs a thorough rewriting for me to be able to understand its contributions clearly and appreciate the same.

**Strengths:**

- None that I could identify.

**Weaknesses:**

- The writing style of the paper is quite informal and unclear. A thorough rewrite is needed for Section 3 to convey the ideas clearly.
- This paper lacks the novelty of the problem definition as well as solution approach. The proposed ideas are well known in the literature.
- Its unclear why standard self-attention would not take care of the QK-score that is being mentioned here.
- Its unclear what and how reasoning is happening in MCQA dataset. No illustrative example is provided.

**Questions:**

- There are some ill formed sentences at multiple places. For example, look at line number 123-124.
- Its unclear why standard self-attention would not take care of the QK-score that is being mentioned here.
- Its unclear what and how reasoning is happening in MCQA dataset. No illustrative example is provided.

---

> ### Author Response · Authors · 2025-11-14
>
> 1. The writing style of the paper is quite informal and unclear. A thorough rewrite is needed for Section 3 to convey the ideas clearly.
>
> **Reply:** We agree with the reviewer that the presentation quality could be further improved. We thank the reviewer for the valuable suggestion to improve Section 3.
>
> 2. This paper lacks the novelty of the problem definition as well as solution approach. The proposed ideas are well known in the literature.
>
> **Reply:** We respectfully disagree with the statement that the proposed ideas are not novel and are already well-known in the literature. We would appreciate it if the reviewer could point us to prior work that questions the novelty of the ideas presented in this manuscript. While we acknowledge that the use of the QK score as a signal was previously known [1] and cited this fact in our paper, it has not been applied to achieve strong results in reasoning tasks at the inference stage. Moreover, the existence of special heads that can be identified through a dedicated selection algorithm and that provide strong reasoning signals via the QK score was not known before.
>
> 3. Its unclear why standard self-attention would not take care of the QK-score that is being mentioned here.
>
> **Reply:** We thank the reviewer for the question, but we are not entirely sure what is meant by the phrase “take care of the QK-score.” We would sincerely appreciate further clarification on this point. If the question refers to the self-attention mechanism that is present in the decoder by default, it is important to highlight that our procedure does not affect the attention calculation process. If, however, the question concerns the type of function that might be used as a signal, we would like to emphasize that we do not claim that better functions than the  QK score product cannot exist. However, for the specific heads identified by our selection procedure, characterized by their particular Query–Key matrices, the QK score on specific tokens provides a meaningful and effective signal. Importantly, if the general self-attention from the model was always a sufficiently qualitative signal, we would not have been able to demonstrate that the QK score signal from specific heads can outperform the baseline.
>
> 4. Its unclear what and how reasoning is happening in MCQA dataset. No illustrative example is provided.
>
> **Reply:** We thank the reviewer for the valuable suggestion that may help improve the presentation quality of our paper. However, we do not fully understand the source of confusion regarding the question “what and how reasoning is happening in MCQA.” We do not modify the reasoning process itself, nor do we require explicit reasoning trajectories. If such trajectories were to be generated, they would be standard. Instead of generating a full reasoning trajectory to select or verify the correct answer, we perform a single forward pass and select the answer based on the QK signal from the pre-selected heads. The complete procedure is described in detail in the present manuscript.
>
> 5. There are some ill formed sentences at multiple places. For example, look at line number 123-124.
>
> **Reply:** We thank the reviewer for noticing this issue and will correct the typographical mistake in the camera-ready version of our paper.
>
> [1]  Tulchinskii, Eduard, et al. "Quantifying Logical Consistency in Transformers via Query-Key Alignment." arXiv preprint arXiv:2502.17017 (2025).

---

> > ### Comment · Reviewer_ZnYX · 2025-11-23
> > **Response to Reviewers' Comments**
> >
> > Thanks for taking time to comment on the review. Below are my responses.
> >
> > As far as novelty is concerned, my major concern is that your proposed method is unclear to me because of the way this section is written currently. I am unable to appreciate the novelty even if there is any. Specifically,
> > 1. There are many ill-formed sentences in Section 3 which makes it hard for me to understand the method details in an unambiguous manner.
> > 2. The finer level details of the proposed method are missing in Section 3. For example,
> >  - You have introduced the generic quantity $S_{QK}^{\ell, h}(c_r, a_r)$.  It was never mentioned what do $\ell$ and $h$ correspond to? I assume $\ell$ corresponds to a layer and $h$ corresponds to an attention head.
> > - I believe this quantity $S_{QK}^{\ell, h}(c_r, a_r)$ is defined for each layer ($\ell$) of the transformer and each attention head ($h$)? If yes, then it is unclear, for say MCQA, how these quantities computed at each layer and head level are combined (or picked one) towards the final decision making in your proposed method. At the end of Section 3, you stated saying _"...we do not aggregate predictions or QK-scores from multiple attention heads. Instead, in each experiment we use a separate calibration subset of the data from the same domain to select the single best performing head..."_. Details of this head selection are missing in this section which is a very crucial step. Suppose I were to implement your proposed method, I have no idea how I shall go about it.
> > - Furthermore, I believe as you go from one layer of transformer to the another layer, the standard multi-head attention calculations take place in usual way and they shape the query and the key vectors at each (layer, head) level which you are consuming in your formula for $S_{QK}^{\ell, h}(c_r, a_r)$? If yes, I am again perplexed what is the new stuff here. Is it just the selection formula for picking the right answer based on the score $S_QK^{\ell, h}(c_r, a_r)$ -- which itself is unclear as I mentioned above.

---

> > > ### Author Response · Authors · 2025-11-24
> > >
> > > We thank the reviewer for the clarifications.
> > > 1. We agree with the reviewer that this section could be further improved, and we have updated it accordingly in the revision.
> > > 2. Yes, both $\ell$ and $h$ correspond to a layer and a head, respectively; we have added a specific clarification in the revision.
> > > 3. We thank the reviewer for raising this concern; we have added the corresponding clarification in the revised version.
> > > 4. We consider the QK score as a valuable interpretability tool that reveals the existence of heads with strong relation to the reasoning process within the model and provides a way to identify them in practice. Also, we show which part of the attention mechanism contains the reasoning "signal”, and demonstrate that it can be efficiently used in various downstream applications. Finally, in all studied tasks, QK-score shows performance on par with that of the model itself and quite often outperforms it, thus showing that it surpasses the general self-attention framework. We believe that our work provides valuable results in interpretability, while also providing potential for some enhancement of the performance.
> > >
> > > P.S. All additions to the text introduced  in the revised version are highlighted in red.

---

> ### Author Response · Authors · 2025-12-03
>
> Taking into account the present circumstances related to the new rebuttal format, and with full respect to the reviewer, we would like to highlight to our AC that the reviewer has not provided meaningful insights to justify the rating they selected. Claims related to low novelty are not supported, and the review itself suggests that the paper was not thoroughly examined. The presentation issues raised by the reviewer have been addressed in our revisions, and we believe that assessing any manuscript in a formal venue solely on presentation quality is both unfair and subjective.
>
> Under standard circumstances, it would be possible to clarify the basis of the rating through a discussion with the reviewer and potentially improve it if no substantial concerns were identified. However, given the current situation, we would like to state that we believe the selected rating is not adequately justified, and we suggest that it should not be taken into account in the final decision.

---

### Official Review · Reviewer_MDeN · 2025-10-26

**Soundness:** 2
**Presentation:** 3
**Contribution:** 3
**Rating:** 4
**Confidence:** 3

**Summary:**

This paper presents a white-box method for answer selection and verification in LLMs. The main idea is to use the raw Query-Key (QK) dot-product score from the transformer's attention mechanism as an "internal signal" from the model. The authors claim that a "think-first" phase via CoT prompting strengthens internal QK alignment, allowing for more reliable selection of answers directly from model activations. This method is evaluated in several settings, such as MCQA, verification, and hypothesis selection. The MCQA setting is tested on the MMLU-Pro and HLE-1/4 datasets; the verification and hypothesis selection settings are tested on MATH-500 and HLE-1/4. HLE-1/4 is an authors' adaptation of the Humanity Last Exam benchmark. Additionally, the MATH-500 dataset is used for testing open-ended reasoning. The authors report significant performance gains of up to ~22% for this method.

**Strengths:**

- Proposed method is novel. While prior work used QK for probing, this paper uses it as a decision rule for selection and verification.
- Some of the reported results are impressive. 1) In the MCQA setup on MMLU-Pro, the QK-score method outperforms the baseline for several models, such as Qwen-14B: 17.72% -> 44.42% or Qwen-32B: 16.6% -> 49.32%. 2) In the hypothesis selection, the QK-score method (tested with LLaMA-3.1 8B on the MATH-500 dataset) outperforms baseline by almost 22 pp.
- Figure 2 shows a high correlation of head performance between MATH-500 and HLE-1/4 for hypothesis selection. This provides evidence that QK-score method captures more generalizable signal.
- I appreciate the usage of the PA metric. Positional bias is often forgotten when using MCQA tasks.

**Weaknesses:**

- The entire method relies on a crucial and potentially fragile step -- selecting a single best-performing head from a calibration dataset. The paper provides no analysis of how stable this selection is. How large does the calibration set need to be? What is the performance distribution across heads? Is there only one good head, or are there many of them? Is head selection done in each setting/experiment separately? If yes, it weakens the claim of generalization of the method. The lack of these analyses is a major limitation of this work.
- The method heavily depends on the choice of "premise-representing" and "response-representing" tokens. The paper lacks an ablation study to show how sensitive are the results to this choice.
- While some results presented are strong, others contradict the paper's main narrative. For example in the MCQA with CoT setting, the QK-score method underperforms the baseline for several models (on HLE-1/4 -> DeepSeek-R1-Distill-Qwen-14B: 33.25% acc on baseline vs. 31.56% acc on QK-score; Qwen3-14B: 33.06% acc on baseline vs. 29.06% acc on QK-score). These underperformances are not discussed or explained, weakening the claim that the QK-score is a systematically better selection mechanism when CoT is used.
- The MCQA baseline in tables 1 and 2 is not explained.

**Questions:**

- How stable is the "golden head" selection? Is the same head selected across different tasks and datasets for a given model?
- Please provide results for a top-k head ensemble to check for robustness, rather than relying on a single head.
- Why does the QK-score method underperform the baseline after CoT is applied in several cases in Table 2? This seems to contradict the main hypothesis.
- Please provide an ablation study on the choice of "premise-representing" and "response-representing" tokens to justify using punctuation.
- How was the "MCQA Baseline" in Tables 1 and 2 calculated?

I am willing to increase my score if these concerns are resolved (especially regarding the head selection).

---

> ### Author Response · Authors · 2025-12-03
> **Rebuttal 1/2**
>
> **General reply:** We thank the reviewer for the reasonable and constructive comments. We have provided the required ablation for the procedure.
>
> The entire method relies on a crucial and potentially fragile step -- selecting a single best-performing head from a calibration dataset. The paper provides no analysis of how stable this selection is. How large does the calibration set need to be? What is the performance distribution across heads? Is there only one good head, or are there many of them? Is head selection done in each setting/experiment separately? If yes, it weakens the claim of generalization of the method. The lack of these analyses is a major limitation of this work.
>
> **Reply:** We thank the reviewer for the questions. Regarding the size of the calibration set:  In our experiments on MMLU_Pro and HLE-¼ datasets we use a calibration subsets of 500 questions in size (section 4.3). Other questions are resolved further.
>
> How stable is the "golden head" selection? Is the same head selected across different tasks and datasets for a given model?
>
> **Reply:** We thank the reviewer for the question, we provide an answer to it in the Hypothesis selection section of our paper with the correlation chart. Performance of QK-heads highly correlates between two benchmarks.
>
> Please provide results for a top-k head ensemble to check for robustness, rather than relying on a single head.
>
> **Reply:** We have obtained the following results for head ensembling (averaging QK-scores from multiple heads for corresponding options) on MMLU_Pro for DeepSeek-R1-Distil-Qwen7B model. MCQA without reasoning:
>
> | Heads Ensembled | Accuracy | Permutation Accuracy |
> |-----------------|----------|-----------------------|
> | Top 1           | 27.29    | 14.92                 |
> | Top 5           | 31.498   | 18.888                |
> | Top 10          | 31.652   | 18.580                |
> | Top 1%          | 31.395   | 18.374                |
> | Top 5%          | 30.983   | 17.807                |
> | Top 10%         | 29.696   | 17.036                |
>
> **MCQA with CoT reasoning:**
>
> | Heads Ensembled | Accuracy | Permutation Accuracy |
> |-----------------|----------|-----------------------|
> | Top 1           | 25.45    | 14.20                 |
> | Top 5           | 24.50    | 13.13                 |
> | Top 10          | 26.59    | 13.13                 |
> | Top 1%          | 25.76    | 12.90                 |
> | Top 5%          | 25.34    | 13.30                 |
> | Top 10%         | 24.82    | 12.70                 |
>
> As we can see, for MCQA without reasoning, aggregation of prediction from several heads leads to certain improvement in performance, while in case of MCQA with CoT reasoning, performance remains nearly the same. In both cases, aggregation of QK-scores fromTop-10 heads
>
>
> Why does the QK-score method underperform the baseline after CoT is applied in several cases in Table 2? This seems to contradict the main hypothesis.
>
>
> **Reply:** We thank the reviewer for the reasonable question. We assume that in some cases the QK signal that is extracted from the specific head might be not enough to completely outperform the model, while we still feel that the same performance is an important finding. Existence of the singular head that will outperform the entire decoder, not just show the same performance might depend on the model size and family. Methods like top-k head ensembling might help in such situation, but it still requires investigation and we assume that it can be done as a future work.
>
>
> Please provide an ablation study on the choice of "premise-representing" and "response-representing" tokens to justify using punctuation.
>
>
> **Reply:**
> We thank the reviewer for the reasonable question. We found out that selection of tokens from which keys are taken is important in our procedure, while selection of the token from which query is taken is less important. We ran our setup 5 times and each time we fixed 4 random tokens from which we take keys and then we aggregated the numbers, while the baseline score was fixed:
>
> **Baseline:** 0.304
> **QK:** 0.264
>
> In the case where keys are taken from random tokens, the QK score significantly underperforms the baseline. However, when we instead select a random token from which to take the query while keeping the key tokens taken from the options as in the original setup the change in the QK score’s performance is insignificant. It is important to note that in this setting we introduced partial determinism: we attempted to select the query from 7 tokens, starting from the last token and moving backward with a step of 2 (i.e., from the second from the end  to the fourteenth from the end). This ensured deterministic positions, while the content of each chosen token was still different.

---

> ### Author Response · Authors · 2025-12-03
> **Rebuttal 2/2**
>
> | **Token Position from End** | **QK Performance** |
> |-----------------------------|---------------------|
> | 2   | 0.33  |
> | 1  | 0.302 |
> | 4  | 0.304 |
> | 6  | 0.304 |
> | 8  | 0.312 |
> | 10 | 0.322 |
> | 12 | 0.31  |
> | 14 | 0.308 |
>
> The general conclusion of the ablation is that the selection of tokens from which key values are taken is crucial for the procedure. At the same time, once the positions for the key tokens are chosen appropriately, the procedure remains stable regardless of the position of the token from which the query is taken.
>
> How was the "MCQA Baseline" in Tables 1 and 2 calculated?
>
> **Reply:** In those experiments, we calculated Baseline according to the standard procedure for MCQA, when the predicted option is chosen as the letter token (‘A’, ‘B’, ‘C’, or ‘D’ for 4-options question) that have the highest log-probability as the first token after the prompt.

---

### Meta-Review · Area_Chair_QLVo · 2026-01-03

**Summary:**

The novelty, experimental fairness, presentation, and clarity issues should be further addressed for future publication. Regarding the formatting issue, the AC has reviewed the submission history and concurs with the reviewers that this paper should have been desk-rejected without further review.

Despite the authors’ rebuttal and subsequent discussion with reviewers, several critical concerns, particularly regarding technical novelty and methodological rigor, remain unresolved. Given ICLR’s exceptionally competitive acceptance rate this year, the AC regrets to recommend rejection.

**Reviewer Concerns:**

The rebuttal addresses several concerns raised. Specifically:

1. MCQA baseline: The authors clarified that the baseline selects the option letter token with the highest log-prob as the first generated token.

2. Top-k head ensembling: The authors provided the head-ensembling results in the table (on one model), showing gains without reasoning and little change with CoT.

3. Token-choice ablation: The authors provided experimental results that key-token choice matters substantially, while query position is less sensitive once keys are chosen appropriately.

Concerns that have not been addressed:

1. Reproducibility / missing experimental details: Reviewers asked for concrete prompts (esp. Sec 4.4/4.5), decoding settings (temperature/top-p/budget), and specified head-selection details. However, the response often says they will add more description.

2. Representation issues: one reviewer could not confidently assess the work due to writing clarity issues, and even after claimed revisions, the AC still found there is a big room for improving the representation of the paper.

3. Novelty: A major issue raised by the reviewer is that MCQA-with-CoT feels incremental over prior QK-based MCQA, and the AC concurs with this point. The rebuttal did not adequately address this issue.

**Reviewer Scores:**

The scores are 4, 2, and 0, and the AC believes the average is unlikely to reach the acceptance threshold given the current author-reviewer discussion.

---

### Decision · Program_Chairs · 2026-01-26

Reject